# Aptamer-Based Detection of Ampicillin in Urine Samples

**DOI:** 10.3390/antibiotics9100655

**Published:** 2020-09-29

**Authors:** Matthew D. Simmons, Lisa M. Miller, Malin O. Sundström, Steven Johnson

**Affiliations:** 1Department of Electronic Engineering, University of York, Heslington, York, North Yorkshire YO10 5DD, UK; mios500@york.ac.uk; 2Department of Chemistry, University of York, Heslington, York, North Yorkshire YO10 5DD, UK; lisa.miller@york.ac.uk

**Keywords:** ampicillin, aptamer, drug-resistant-infections, antimicrobial resistance, diagnostics, gold nanoparticles, beta-lactamase, beta-lactam, fluorescence, urine analysis

## Abstract

The misuse of antibiotics in health care has led to increasing levels of drug resistant infections (DRI’s) occurring in the general population. Most technologies developed for the detection of DRI’s typically focus on phenotyping or genotyping bacterial resistance rather than on the underlying cause and spread of DRI’s; namely the misuse of antibiotics. An aptameric based assay has been developed for the monitoring of ampicillin in urine samples, for use in determining optimal antibiotic dosage and monitoring patient compliance with treatment. The fluorescently labelled aptamers were shown to perform optimally at pH 7, ideal for buffered clinical urine samples, with limits of detection as low as 20.6 nM, allowing for determination of ampicillin in urine in the clinically relevant range of concentrations (100 nM to 100 µM). As the assay requires incubation for only 1 h with a small sample volume, 50 to 150 µL, the test would fit within current healthcare pathways, simplifying the adoption of the technology.

## 1. Introduction

Antibiotics are used extensively for the vital treatment of bacterial infections in both humans and animals. The misuse and overreliance upon antibiotics has accelerated the development of antimicrobial resistance (AMR) against a large variety of commonly used antibiotics [1]. This in turn had led to a clinical increase in difficult-to-treat drug-resistant infections (DRI’s) for multiple bacterial strains [1,2,3].

β-Lactam-based antimicrobials are among the most commonly used classes of antibiotics, showing effectiveness against a wide variety of both Gram-positive and Gram-negative bacteria [4,5,6]. However, overuse of this important class of antibiotics, partly due to poor dosage management and patients not complying with treatment, has led to the emergence of high levels of resistance [7]. For example, it has been suggested that up to 75% of critically ill patients in intensive care may not be receiving appropriate doses of β-lactam class antibiotics [8] and in excess of 95% of pathogenic *Staphylococcus aureus* worldwide is resistant to β-lactam antibiotics [7]. This high prevalence of resistance has dramatically restricted the use of previously common β-lactam antibiotics. Ampicillin, for example, was the most commonly used β-lactam class antibiotic in healthcare but increasingly, pathogenic *Escherichia coli* strains are showing resistance to ampicillin, decreasing effectiveness in the treatment of common infections. The misuse and non-optimal dosage of antibiotics has therefore led to an increase in recurrent urinary tract infections (UTI’s), with a 16.1% increase in cases of drug resistant UTI’s between 1990 and 2013 globally leading to over >93 million cases worldwide and 10.5 million visits to physicians in the US alone [8,9].

Current techniques to determine resistance rely on culturing of bacterial strains in the presence of antibiotics. While such susceptibility tests can be very accurate, they are typically slow, taking up to 48 h [8] to achieve a definitive result. To combat the rise in DRI’s, a wide variety of tests are being developed for the purpose of determining the presence of antimicrobial resistant pathogens in shorter time frames and often focus on the development of new technologies [10,11,12,13] for point-of-care diagnostics including: PCR [14,15], nucleic acid amplification techniques (NAAT) [16], rapid antibiotic susceptibility testing [17,18], electrochemical detection [19] and SERS [20,21,22]. Several reviews have also been published on this topic examining rapid point-of-care tests and their adoption for both respiratory and urinary tract infections and their adoption into clinical pathways [11,23,24,25,26]. While these technologies show great promise, they often require current clinical pathways within healthcare systems to be adapted, or are deployed ineffectively, limiting their uptake [5,27,28,29,30,31,32]. Rapid adoption of new diagnostics would thus ideally employ equipment and processes commonly found in hospital laboratories. Furthermore, despite the importance of appropriate dosage management and patient compliance in tackling the spread and emergence of resistance [33], the majority of these emerging technologies focus instead on the direct detection of DRI’s and thus only partly address the misuse of β-lactam antibiotics.

It has been suggested that a more effective approach to tackling AMR and the DRI’s caused by this is to prevent the misuse of antibiotics in the first place. To this end, high performance liquid chromatography (HPLC) methods in combination with various forms of detection have been applied for determination of antibiotic concentrations in plasma samples [34,35,36]. While accurate, enabling quantification on the nM to µM range on a short time scale (typically less than 20 min), these techniques require invasive sampling to collect and process the blood plasma as well as expensive equipment and expert users [37].

Diagnostic assays based on fluorescently labelled aptamers have the potential to satisfy the needs for rapid detection and compatibility with established laboratory protocols, typically only requiring a plate reader for concentration read out. DNA aptamers are single stranded DNA (ssDNA) molecules that are capable of binding a wide range of biomarkers including microbial and viral pathogens [38,39,40,41,42,43], and small molecules [43]. The high affinity and specificity of aptamers relates to their capability to recognise a target molecule through interactions between the 3D conformation of the folded aptamer and the target [44]. Given the high binding affinity and selectivity coupled with in vitro selection and low-cost synthesis, they are ideal candidates for use in a variety of healthcare diagnostics. Furthermore, they are compatible with a wide variety of readout systems including time resolved fluorescence [45], magnetic resonance imaging [46,47], flow cytometry [38,48], electrochemical sensors [39,42,49], and lateral flow devices [41,43,50].

Recently, an aptameric based assay has been developed by Song et al. for the quantification of ampicillin in heavily processed milk in the agricultural industry using the first aptamer reported for ampicillin detection [4]. The assay combined fluorescently-labelled aptamers specific to ampicillin that was non-specifically bound to gold nanoparticles (AuNPs) that acted as a quencher for the fluorescent tag. In the presence of ampicillin, the aptamer dissociates from the AuNPs, resulting in a fluorescent readout and subsequent aggregation of the gold nanoparticles, Figure 1. The assay was demonstrated using processed milk as a sample matrix, that was buffered to pH 8 where the signal of the fluorescent tag was optimal and diluted to minimise potentially interfering components in the milk that otherwise induced nanoparticle aggregation. While the assay is suitable for use in an agricultural setting, the requirement to heavily dilute and buffer the sample would limit its use and its adaptability for use in other settings.

Here we show the development of an aptamer-based assay previously used in a non-clinical setting and developing this for the optimisation of ampicillin dosage and compliance by monitoring a patient’s urine sample. Specifically, we demonstrate a diagnostic assay based on fluorescently-labelled aptamers that is capable of quantifying the concentration of ampicillin in clinical urine samples with a high degree of sensitivity at clinical levels. Readout and quantification of the assay is achieved using a standard laboratory fluorometer, requires limited pre-sample preparation and is thus suitable for translation into established clinical pathways to support the informed use of this valuable antibiotic [4].

## 2. Results

### 2.1. Determination of Ampicillin in Buffer

The aptamer used in this work was previously reported by Song et al. [4]. The ssDNA aptamer (3′-GCGGGCGGTTGTATAGCGG-5′) was selected against ampicillin via the SELEX technique and was confirmed to bind to ampicillin with a high degree of sensitivity and specificity (K_d_ = 13.4 nM). Initial testing of the aptamer sequence for use in clinical urine samples was performed here through the introduction of the fluorescent fluorescein derivative, FAM, at the 5′-end of the oligonucleotide (3′-GCGGGCGGTTGTATAGCGG-5′-FAM). Urine samples were collected anonymously from 13 healthy adult volunteers that had not shown symptoms of infection and had not taken antibiotics within 1 month prior to the sample collection. Participants were recruited from within the University of York. Samples were filter sterilised using 0.22 μm syringe filters to remove any cells. The urine samples were combined to form an “average” human urine pool, which was aliquoted and stored in a −80 °C freezer. The pH of the resulting urine was measured to be pH 6.7. All collection and processing of the samples was performed and conforms to the University of York’s Ethical approval and clinical sample handling guidelines.

The labelled aptamer strand (10 nM) was annealed at 95 °C for 5 min in potassium phosphate buffer containing magnesium chloride (KPi, 100 mM + 1 mM MgCl_2_, pH 8), before cooling to 4 °C. The annealed aptamer strands were subsequently incubated with AuNP’s, producing an AuNP-aptamer conjugate with a quenched fluorescence. Ampicillin samples of known concentrations were prepared by serial dilution of a 10 mM stock dissolved in KPi buffer (pH 8). Samples containing ampicillin (1 nM to 1 mM) were subsequently mixed with the AuNP-aptamer conjugate in triplicate and incubated for a further 1 h at room temperature in a 96-well plate. The fluorescent signal (λ_ex_ = 487 nm, λ_em_ = 528 nm) of the solutions was subsequently measured relative to the fluorescence intensity of a blank sample of buffer with no ampicillin present. The continuous band structure of the AuNPs effectively quenches the fluorescence of the fluorescent tag when the aptamer is bound passively to its surface. Upon incubation with ampicillin the aptamer selectively binds to the antibiotic, dissociating from the surface of the AuNPs, allowing the fluorescence of the FAM tag to be quantified and calibrated against the concentration of ampicillin present in the sample. An increase in the fluorescence was therefore seen with the increasing concentration of ampicillin present in the buffer, as shown in Figure 2. We note, the high baseline level of fluorescence is believed to be due to incomplete quenching of the FAM-labelled aptamers. This could be improved by further assay optimisation, particularly the relative aptamer: AuNP ratio.

Based on previous methods to quantify the sensitivity of diagnostic assays, [51,52] we have employed both the limit of the blank (LoB)—the minimum concentration that can be detected above a blank sample containing no ampicillin, and the limit of detection (LoD)—the lowest concentration of ampicillin that can be measured with statistical confidence. The LoB was determined to be 1.55 µM. The LoD, based upon the minimum concentration measured plus a single standard deviation, was slightly higher at 3.75 µM for the unoptimised assay in buffer. While the assay is therefore responsive to ampicillin concentration, its application as a clinical assay where concentrations of ampicillin in urine range from 1 µM to 1 mM, would be limited. The high limits of detection here are likely due to inaccuracy at the lower end of the concentration range tested, and the larger variance seen in the sample value, particularly at 10 nM concentrations.

The selectivity of the aptamer was also analysed here against the β-lactam antibiotics cephalexin and amoxicillin, which are structurally similar to ampicillin. The enzymatically hydrolysed form of ampicillin was also compared, by first exposing ampicillin to a blend of β-lactamase enzymes, the chemical structures of which can be seen in Figure 3. These measurements allowed for the determination of whether the aptamer was selective for the overall molecular structure of ampicillin, rather than the β-lactam ring structure or hydrolysed by-products. Solutions of cephalexin, amoxicillin, and hydrolysed ampicillin (all at 1 mM) were prepared in KPi buffer and incubated with the preformed AuNP-aptamer conjugate. As shown in the inset of Figure 2, a greater fluorescence signal, almost double that of cephalexin, amoxicillin, or hydrolysed ampicillin, was obtained in the presence of ampicillin. The aptamer therefore shows selectivity for the intact ampicillin over the structurally similar antibiotics (cephalexin and ampicillin) or the hydrolysed form of ampicillin.

Control assays were also performed using either only AuNP’s and varying concentrations of ampicillin, or a non-selective ssDNA labelled with FAM (3′-TATGCGCCGGTTTTCAGCCT-5′-FAM) incubated with AuNP’s and the varying concentrations of ampicillin, as per the assay. Negligible fluorescence was observed for the blank assay without FAM labelled aptamer. When the non-selective FAM labelled ssDNA strand was used instead of the aptamer, we observed no increase in fluorescence with increasing concentrations of ampicillin (Appendix A
Appendix A).

### 2.2. FAM Labelled Aptamers for Use in Urine Based Assays

The pH of urine from healthy individuals can typically vary between pH 6 to 7. However, for certain illnesses, including UTIs, the urine pH can vary between pH 4.5 to 8 [53]. The overall pH of the urine sample was not expected to directly influence hydrolysis of the ampicillin, as over a short time period (<10 days) the rate at which hydrolysis of ampicillin occurs has been shown to be independent of pH, only the hydrolysis product created changes [54]. The fluoresence of fluorescein derivative labels, such as FAM, however, are known to be affected by changes in pH [55]. Fluorescein itself has a pKa value of 6.4 and is only recommended for use in a pH range between 5–7 [56]. Clinical urine samples are typically collected and stored in tubes coated with boric acid (2% v/v) as a preservative to maintain the sample prior to testing [57]. Critically, the boric acid also buffers the urine typically within the range pH 6–8. To determine the efficacy of the FAM-labelled ampicillin aptamer assay over the expected clinical range, dilutions of ampicillin in KPi buffer at pH 6–8 were prepared. The salt molarity of the solution was maintained so as not to affect the isoelectric point of the AuNP and interfere with the passive binding and desorption of the aptamer.

After the passive adsorption of the aptamer to the AuNPs, the aptamer-AuNP conjugates were immediately incubated with ampicillin in KPi buffered at pH 6–8. As shown in Figure 4, the sensitivity of the assay is optimal at pH 7, showing greater fluorescence intensities at lower concentrations, and having a greater gradient of the calibration curve in the desired concentration range (100 nM to 1 mM). As expected with Fluorescein derivatives, the fluorescence intensity and gradient decrease at pH 8, however the slope and intensities would still be sufficient for determination of the ampicillin concentration in buffer at the desired clinical concentrations. Conversely, at pH 6 we observed a significant reduction in fluorescence intensity, and the gradient of the curve was much shallower, limiting the accuracy of the test at acidic pH. The LoDs determined from the lowest concentration measured were found to be 9.16 µM, 2.18 µM, and 11.7 µM for pH 6, 7 and 8, respectively. While the curve gradient at pH 8 would allow it to function within the detection limit as a sensor, the LoD is compromised. For use in a clinical setting, the assay would need to contain a buffering agent to help buffer the pH of the assay after sample addition to an optimal pH 7, necessitating an additional processing step over that typically used clinically for urine sample collection.

As with the initial attempt to determine ampicillin concentrations using the FAM labelled aptamer strand, the higher than desired limits of detection are due to variance at the lower end of the concentration range tested. This variance may not only occur due to the pH fluctuation, but also the interference of ampicillin and its hydrolysed form which occurs at similar wavelengths to that of the FAM fluorescent tag, as discussed below and shown in Figure 5.

### 2.3. Development of Test to Determine Ampicillin Concentration in Urine

Development of a test that works in unprocessed urine samples necessitates a change of the fluorescent label not only due to the pH sensitivity of the label but also the intrinsic fluorescence exhibited by urine. This intrinsic fluorescence significantly overlaps that of the fluorescein based fluorescent tags used in the previous work. The fluorescence of urine is due to the presence of flavins, which are normally found in urine at low concentrations in healthy patients and exhibit a fluorescence emission at approximately 520 nm [58]. As shown in Figure 5, the fluorescence of urine overlaps spectrally with that of FAM but not Alexafluor647. The problem of spectral overlap is further exacerbated by the presence of any hydrolysed ampicillin within the sample. While the aptamer-based assay itself demonstrates some selectivity between hydrolysed and non-hydrolysed ampicillin, both ampicillin and its hydrolysed products display an intrinsic fluorescence, which—depending on the concentration—can interfere with that of the FAM label. β-Lactam antibiotics are readily hydrolysed by β-lactamase enzymes, which are one of the main modes of resistance in drug resistant UTI’s [5,59]. β-Lactamase enzymes catalyse the hydrolysis of the β-lactam ring structure resulting in the deactivation of the antibiotic, and a change in the fluorescence spectrum.

Emission/excitation fluorescence spectra of the aptamer strands labelled with Alexafluor647 in undiluted urine samples were compared to those of unprocessed urine, FAM labelled aptamer, ampicillin, and hydrolysed ampicillin at the required concentrations to identify regions of spectral overlap. Hydrolysed ampicillin was synthesised by exposure of a stock solution in KPi buffer at pH 7 to a blend of β-lactamase enzymes that was incubated at 37 °C for 48 h. The β-lactamase enzymes were subsequently filtered from the samples under centrifugation with a 30 KDa filter and the hydrolysed ampicillin collected and analysed using ^1^H NMR and mass spectroscopy to determine the degree of hydrolysis (Appendix A
Appendix A). After analysis, the resulting stock of hydrolysed ampicillin was diluted to the required concentrations.

The emission excitation spectra of the FAM and Alexafluor647 bound aptamers, have characteristically narrow emission spectra. Specifically, FAM fluorescence was found to occur between 360–400 nm when excited at 350 nm, and between 520–540 nm when excited at 450–500 nm. Similarly, Alexafluor647 fluoresced at a single emission peak between 660–700 nm when excited at either 525 or 600–650 nm. This is in good agreement with the fluorescence spectra expected for these probes [56].

Urine samples used in this paper possess a broad fluorescence band between 350–600 nm when excited at wavelengths between 300–500 nm. This overlaps directly with the fluorescent spectrum of FAM. While ampicillin shows a broad fluorescence between 390 and 530 nm when excited at wavelengths below 400 nm, upon hydrolysis this fluorescence shifts to longer wavelengths between 475 and 550 nm when excited at 425 nm, which was not present in the ampicillin spectra. The fluorescence signals of Alexafluor647 are quite distinct and separate from any of the other fluorescent reagents present within the sample matrix. 2D emission excitation spectra are shown in the Appendix A
Appendix A.

Having established that the fluorescence spectra of Alexafluor647 can be distinguished from competing fluorescent components within the assay and sample matrix, the sensitivity of the assay using Alexafluor647-labelled ampicillin aptamer was assessed. Again, a dilution series of ampicillin was prepared between 1 mM and 1 nM in both KPi Buffer (pH 7) and in filter-sterilised urine samples. The aptamer-AuNP conjugate was incubated for 1 h at room temperature prior to addition of the ampicillin samples which was then incubated for a further hour at room temperature, before the fluorescence of the Alexafluor647 tag (at λ_Ex_ = 640 nm, λ_Em_ = 681 nm) was quantified. The calibration curve for the Alexafluor647-labelled ampicillin aptamer is shown in Figure 6.

Both the buffer and urine assay results show the standard assay response curve expected for biochemical based assays, with the gradient of the slope maximised over the expected clinical range for ampicillin concentrations in urine (100 nM to 100 µM) [60]. The urine-based assay shows only a slight drop in fluorescence emission compared to that performed in buffer, presumably due to scattering of the light in the urine matrix. The LoD, calculated from the minimum concentration of ampicillin measured and its standard deviation, were found to be 43.8 nM and 20.6 nM for the buffer and urine sample matrices, respectively. These LoD values suggest that the assay is suitable for use over the desired clinical range. Typically, the maximum concentration of antibiotics in human urine is around 80 to 115 μg/mL (236 to 322 µM), dependent on the age and health of the patient [60]. The assay would further prove suitable for monitoring of ampicillin levels used in veterinary practices where ampicillin levels in urine excretions have been measured to be 309 ± 55 μg/mL (approx. 0.8 mM) after 8 h in dogs [61]. Control assays can be found in the Appendix A.

The short incubation time of 1 h coupled with the compatibility of the assay with conventional 96-well format fluorescence measurements would allow the technique to easily fit into current protocols employed in both healthcare and veterinary practice, without the need for specialised equipment or training.

## 3. Discussion

An aptameric-based assay has been demonstrated for the detection and quantification of ampicillin in urine, using a fluorescently labelled AuNP-aptamer conjugate. By optimising the AuNP: aptamer concentration ratio and fluorescent label, the assay was shown to be less sensitive to interference from the sample matrix and/or any hydrolysed β-lactams present within the clinical sample. The concentration of ampicillin could be determined down to 43.8 nM and 20.6 nM in buffer and urine, respectively. The determination of ampicillin concentrations to these levels meets those that would be required for determination of ampicillin concentrations in clinical samples, which typically lie in the µM range, depending on dose and excretion rate. Using this assay, the ampicillin concentration can be determined within an hour of sample incubation; thus, it would be possible to use this test to allow for the determination of optimal dosage and monitoring patients’ compliance with prescriptions.

The stability of the aptamer typically led to the assay being used within 28 days. However, long-term stability studies are not within the scope of this work. After this time, it was seen that reproducibility began to suffer and variance in the results increased, suggesting that the fluorescently-labelled aptamer was not stable beyond this point.

As shown in Table 1, the aptamer-based assay presented here only requires a standard plate and provides an effective way of measuring ampicillin concentration in urine samples, without any prior sample processing. Whereas both HPLC and SERS analyses require specialised and expensive equipment and usually a dedicated experienced operator.

## 4. Materials and Methods

Ampicillin was purchased from Sigma-Aldrich (A1593), cephalexin monohydrate was purchased from Fluorochem (463365), and amoxicillin trihydrate purchased from Apollo Scientific (BIA0101), and stored at 8 °C.

β-lactamase enzymes were purchased from Sigma-Aldrich (L7920), as a blend of recombinant proteins, expressed in *E. coli*, which contains two individual β-lactamases (β-lactamase I (600–1500 IU per vial) and β-lactamase II (60–150 IU per vial). β-lactamases were dissolved in KPi buffer (50 mM) and stored in aliquots of β-lactamase I (60–1500 IU/mL) and β-lactamase II (6–15 IU/mL) at −20 °C.

The ampicillin specific DNA aptamer (3′-GCGGGCGGTTGTATAGCGG-5′) and control aptamers (3′-TATGCGCCGGTTTTCAGCCT-5′-FAM) with their respective fluorescent labels were custom synthesized by and purchased from Thermo Fisher Scientific (Life Technologies). The aptamer was supplied modified with either a FAM or Alexafluor647 fluorescent tag modification to the 5′ end. The lyophilised aptamers were dissolved to the required stock concentration using KPi buffer (100 mM, pH 8) containing magnesium chloride (MgCl_2_, 10 mM) and stored at 8 °C. The fluorescently labelled aptamers, both the FAM and Alexa647 derivatives, were seen to be stable for over 28 days when dissolved and aliquoted in KPi buffer (100 mM, pH 8) containing magnesium chloride (MgCl2, 10 mM) and stored at 8 °C.

The AuNPs were synthesised according to the Turkevich method, using sodium citrate to reduce HAuCl_4_ [63]. All chemicals were purchased from Sigma-Aldrich unless otherwise stated. A flask containing AuClH_4_ (0.2% *w/v*, 7 mL) MQ water (43 mL) was heated to 110 °C. Once the reaction solution was boiling, trisodium citrate (6 mL of 1% *w/v*) was added. After 10 min the reaction mixture had turned from colorless to red. The flask was then removed from the heating block and allowed to return to room temperature. Analysis by UV-vis spectrometry determined the resulting solution to be OD 9 with (λmax = 527 nm). The solution was further diluted with MQ water to give a final solution of OD 0.9 and stored at 4 °C. The average particle diameter of the AuNPs was 13 nm, as determined by UV/Vis Spectroscopy [64].

Urine samples were collected anonymously from 13 healthy adult volunteers that had not shown symptoms of infection and had not taken antibiotics within 1 month prior to the sample collection. Participants were recruited from within the University of York. Samples were filter sterilised using 0.22 μm syringe filters to remove any cells. The urine samples were combined to form an “average” human urine pool which was aliquoted and stored in a −80 °C freezer. The pH of the resulting urine was measured to be pH 6.7 All collection and processing of the samples was performed and conforms to the University of York’s Ethical approval and clinical sample handling guidelines.

### 4.1. Preparation of β-Lactam Antibiotic Solutions

β-lactam antibiotics (ampicillin, cephalexin, and amoxicillin) were prepared directly before use by dissolving in KPi buffer (100 mM) to a stock concentration of 10 mM and stored at 8 °C until used. For use in the assay, serial dilutions of the stock concentration of aptamer were prepared in KPi buffer (100 mM) at the required pH, between 1 nM and 10 mM.

### 4.2. Preparation of Hydrolysed β-Lactam Solutions

Hydrolysed forms of the β-lactam antibiotics (ampicillin, cephalexin, and amoxicillin) were prepared by dissolving the drug in KPi buffer (100 mM, pH 7) to form stock solutions (10 mM). To 1 mL of antibiotic stock solution was added 100 μL β-lactamase blend (β-lactamase I (60–1500 IU/mL) and β-lactamase II (6–15 IU/mL)) in KPi buffer (50 mM, pH 7). The solutions were then vortexed and incubated for 48 h at 37 °C. β-lactamase enzymes were then removed from the solution using a centrifugal filter (Amicon^®^ Ultra 0.5 mL, Millipore (UK) Limited, Watford, UK) with a 30 KDa cut-off and the hydrolysed β-lactam recovered. Hydrolysis of the β-lactam was confirmed by high resolution mass spectrometry and ^1^H NMR.

For use in the assay, serial dilutions of the stock concentration of the hydrolysed β-lactam antibiotics (10 mM) were prepared in KPi buffer (100 mM) at the required pH between 1 nM and 10 mM.

### 4.3. Determination of Ampicillin Concentrations in Buffer and Urine Using Aptamer Based Assay

Prior to use, the aptamers were denatured at 95 °C for 5 min before cooling to 4 °C, in 200 µL aliquots of annealing buffer (100 mM KPi, 1 mM MgCl_2_, pH 8. The fluorescently labelled aptamer was subsequently diluted from the stock to the desired concentration (10 nM and 100 nM for the FAM and Alexafluor647 labelled aptamer strands, respectively) in KPi buffer (100 mM, pH 7) before a 50 μL aliquot was incubated with the AuNP solution (Ø ≈ 13 nm, OD 0.9, 50 µL). Ampicillin in KPi buffer (100 mM, pH 7) was prepared at various concentrations between 1 mM and 1 nM by serial dilution before a 50 µL aliquot was mixed with the DNA-aptamer solution and incubated for 1 h at room temperature.

Fluorescence emission was recorded for each sample (each concentration analysed in triplicate) using a Synergy H1 microplate reader (Biotek, UK). FAM labelled aptamers were measured at λ_ex_ = 487 nm, λ_em_ = 528 nm while the emission intensity of the Alexafluor647 labelled aptamers was measured at λ_ex_ = 640 nm, λ_em_ = 681 nm.

### 4.4. Emission/Excitation Fluorescence Spectra

Fluorescence spectra of all β-lactam antibiotics (10 mM), and their hydrolysed forms (10 mM), as well as urine, KPi Buffer (100 mM), AuNPs (OD 0.9) and fluorescently tagged aptamer ssDNA (100 nM), were collected using a Synergy H1 microplate reader (Biotek, UK). Then, 100 µL of each reagent solution was dispensed into a 96-well plate at the concentrations used in the assay. Emission spectra were collected between λ_em_ = 300–700 nm at 10 nm intervals, the excitation wavelength was varied between λ_ex_ =300–700 nm at 25 nm intervals. Excitation spectra were collected between λ_ex_ = 300–700 nm at 10 nm intervals, the emission wavelength was varied between λ_em_ = 300–700 nm at 25 nm intervals.

### 4.5. Preparation and Determination of Control Assays

The blank control assay, where no fluorescently labelled aptamer was performed by, taking 200 µL aliquots of annealing buffer (100 mM KPi, 1 mM MgCl_2_, pH 8) and heating this to 95 °C for 5 min before cooling to 4 °C. This buffer was then incubated with the AuNP solution (Ø ≈ 13 nm, OD 0.9, 50 µL), for 1 h at room temperature. Ampicillin in KPi buffer (100 mM, pH 7) was prepared at various concentrations between 1 mM and 1 nM by serial dilution before a 50 µL aliquot was mixed with the DNA-aptamer solution and incubated for 1 h at room temperature. This heating and cooling cycle is standard and is included to remove misfolds in the aptamer structure [65].

Fluorescence emission was recorded for each sample (each concentration analysed in triplicate) using a Synergy H1 microplate reader (Biotek, UK). FAM labelled aptamers were measured at λ_ex_ = 487 nm, λ_em_ = 528 nm while the emission intensity of the Alexafluor647-labelled aptamers was measured at λ_ex_ = 640 nm, λ_em_= 681 nm.

For the control assay where a non-selective ssDNA strand labelled with FAM (3′-TATGCGCCGGTTTTCAGCCT-5′-FAM) was used in place of the selective aptamer strand labelled with FAM, the following procedure was used. Prior to use, the control aptamer was denatured at 95 °C for 5 min before cooling to 4 °C, in 200 µL aliquots of annealing buffer (100 mM KPi, 1 mM MgCl_2_, pH 8). The fluorescently labelled aptamer (3′-TATGCGCCGGTTTTCAGCCT-5′-FAM) was subsequently diluted from the stock to the desired concentration (100 nM) in KPi buffer (100 mM, pH 7) before a 50 μL aliquot was incubated with the AuNP solution (Ø ≈ 13 nm, OD 0.9, 50 µL). Ampicillin in KPi buffer (100 mM, pH 7) was prepared at various concentrations between 1 mM and 1 nM by serial dilution before a 50 µL aliquot was mixed with the DNA-aptamer solution and incubated for 1 h at room temperature.

Fluorescence emission was recorded for each sample (each concentration analysed in triplicate) using a Synergy H1 microplate reader (BioTek UK, Swindon, UK). FAM labelled aptamers were measured at λ_ex_ = 487nm, λ_em_ = 528 nm.

## 5. Conclusions

In conclusion, the assay developed in this work can determine the concentration of the β-lactam samples to as low as 20.6 nM directly from urine samples through the use of fluorescently-labelled aptamer and standard plate reader within 1 h of sample addition, without the need for expensive equipment or specialised personnel to interpret the results. By using a simple fluorescent assay, the technique aligns well with conventional healthcare pathways within a clinical setting, rather than requiring the development of new processes and equipment. These benefits would in turn allow for more effective antibiotic use, helping to overcome the development of drug-resistant infections.

## Figures and Tables

**Figure 1 antibiotics-09-00655-f001:**
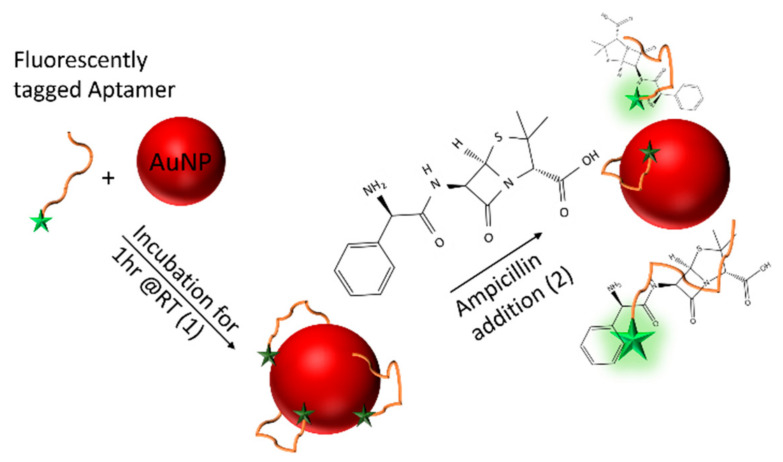
Schematic diagram of the ampicillin aptamer assay. The fluorescently-labelled aptamer is initially bound, non-specifically to gold nanoparticles (AuNPs), leading to quenching of the fluorescence label (1). In the presence of ampicillin, the aptamer dissociates from the AuNP surface in order to selectively bind to the drug leading to an increase in the fluorescent signal (2). The intensity of the fluorescence is proportional to the concentration of ampicillin present within the sample.

**Figure 2 antibiotics-09-00655-f002:**
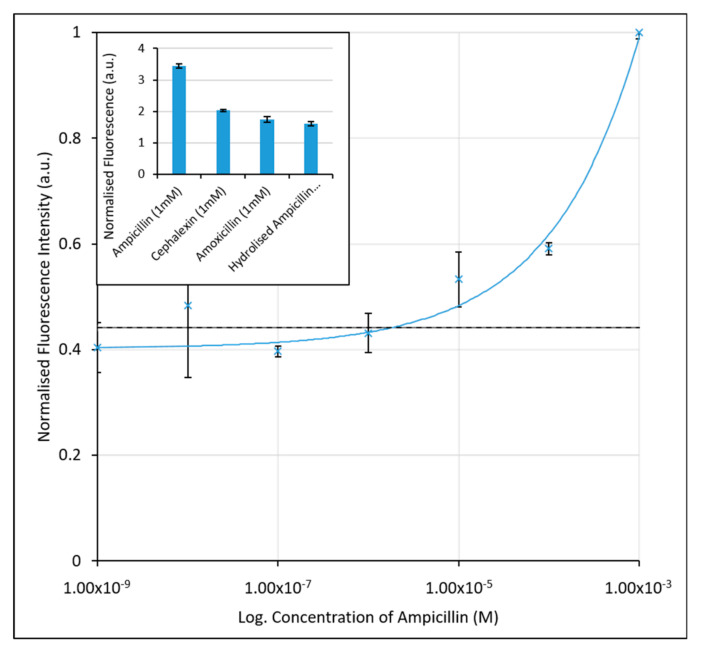
Calibration graph for the determination of ampicillin in KPi Buffer using the initial FAM-labelled aptamer assay. An increase in fluorescence is seen with increasing ampicillin concentration due to desorption of aptamers from the nanoparticle surface. The black, dashed line (at 3.75 µM) corresponds to the limit of detection for the assay. Inset: Fluorescence intensities (λ_ex_ = 487 nm and λ_em_ = 528 nm) of ampicillin, cephalexin and amoxicillin samples at 1 mM concentrations, indicating the aptamers selectivity for ampicillin over similarly structured β-lactamase molecules, and the hydrolysed form of ampicillin. The error bars represent a single standard deviation from the mean. R^2^ value of fitted line is 0.9848.

**Figure 3 antibiotics-09-00655-f003:**
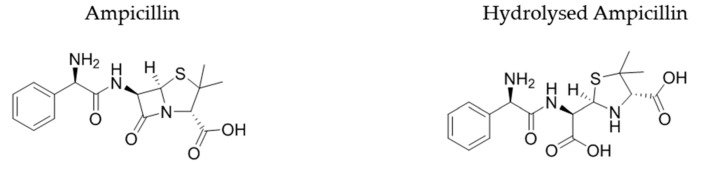
The chemical structures of ampicillin and its hydrolysed form resulting from the enzymatic cleavage of the β-lactam ring structure.

**Figure 4 antibiotics-09-00655-f004:**
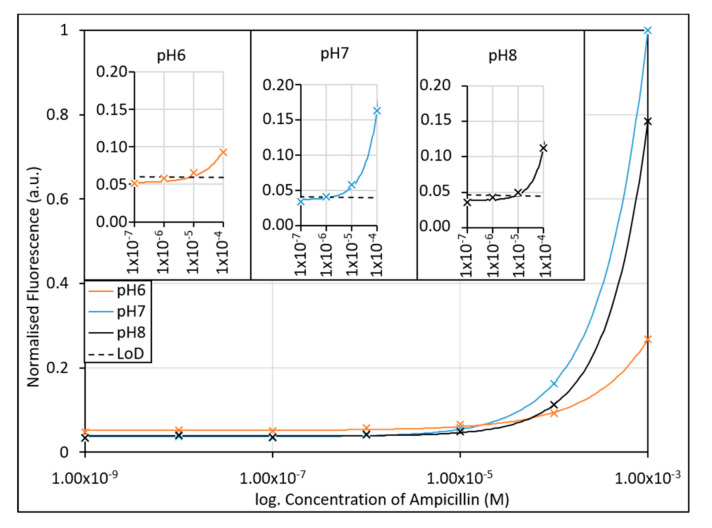
The effect of pH on the ampicillin aptamer assay was investigated using potassium phosphate buffers with pH’s over the expected clinical range (pH 6–8). Due to the fluorescence tag employed, the assays were expected to respond most effectively at neutral pH (pH 7), with a decrease in response to the presence of ampicillin seen at pH 8 and pH 6. Inset: Limits of detection of the assay when performed in different pH buffers. The lowest LoD (2.18 µM) was found to occur at pH 7. R^2^ values of fitted lines are 0.9984, 0.9999, and 0.9999 for pH6, pH7 and pH8, respectively.

**Figure 5 antibiotics-09-00655-f005:**
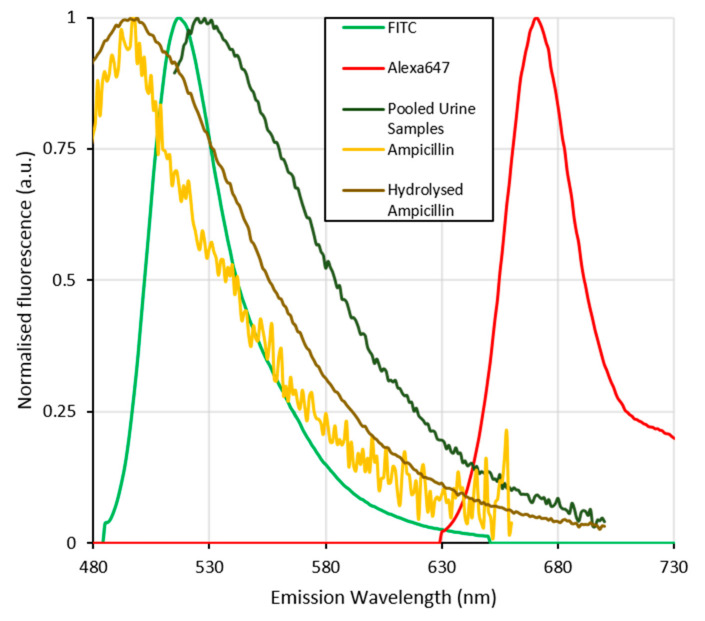
Normalised fluorescence emission spectra of the fluorescent dyes Alexafluor647 and FITC (analogous with fluorescine (FAM)) used to label the aptamer strands compared with those of the pooled urine samples, ampicillin and its hydrolysed form. The spectral overlap between the inherent fluorescence of urine, ampicillin and its hydrolysed form with that of the FAM labelled aptamer can be seen to seriously interfere with the signal from the FAM dye. This and the previously shown pH stability of the flourescine based dyes further complicates the use of this dye in detecting ampicillin in the pooled urine samples. The fluorescence of Alexafluor647 labelled aptamers is significantly separated from that of the matrix components (urine, ampicillin and hydrolysed ampicillin) and thus more suitable for the determination of ampicillin concentrations in clinical urine samples.

**Figure 6 antibiotics-09-00655-f006:**
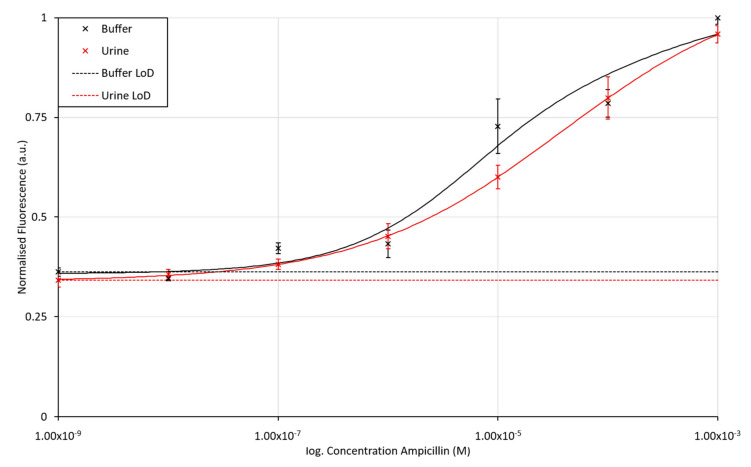
Fluorescence intensity for the Alexafluor647-labelled ampicillin aptamer assay when exposed to varying concentrations of ampicillin in buffer and urine matrices. The LoD for ampicillin in KPi buffer and unprocessed and unbuffered urine was found to be 43.8 nM and 20.6 nM, respectively. The error bars in the figure represent a single standard deviation from the mean. R^2^ values for fitted lines are 0.9674 and 0.9999 for the assay performed in buffer and urine, respectively.

**Table 1 antibiotics-09-00655-t001:** Comparison of various techniques for the detection of β-lactam class antibiotics.

Technique	Antibiotic	Sample Matrix	Time Taken	LoD (µg/mL)	LoD (µM)	%cv (%)	[Ref]
Dip-stick inhibition	Amoxycillin	Urine	1 to 14 days	N/A ‡	N/A	N/A	[62]
HPLC with UV-Vis detection	Various β-lactams Ampicillin	Plasma	1 to 25 min *	4.0–197 4.8–130	11.4–563.81 3.7–372.1	0.4–6.8 0.7–6.5	[34]
UHPLC with UV-Vis detection	Ampicillin	Plasma	8 min *	0.5	1.4	0.03–1.8	[35]
HPLC-MS	Ampicillin	Plasma	13 min *	0.05–50	0.143–143.1	N/A	[37,36]
Surface Enhanced Raman Spectroscopy (SERS)	Ampicillin	DI Water	n/a	0.027	0.07	N/A	[20,21]
Surface Enhanced Raman Spectroscopy (SERS)	Ampicillin	Buffer	20 min *	349	1000	N/A	[20,22]
Fluorescently labelled aptamer detection	Ampicillin	Urine	60 min	0.007	0.026	1–9%	

‡ indicates that the test uses an inhibition technique to determine the effectiveness of the antibiotic and therefore does not give a concentration readout. * Indicates the time shown is the run time of the technique only not including the additional sample preparation time.

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
