# Peer review of "Aptamer-Based Detection of Ampicillin in Urine Samples"

_antibiotics, 2020, doi:10.3390/antibiotics9100655_

Round 1

Reviewer 1 Report

Overall I enjoy reading the article, the article is about rapid and sensitive detection of ampicillin antibiotics in urine samples. The authors considered carefully regarding other ampicillin-like antibiotics and enzyme break down ampicillin products to evaluate if significant different signal level can be obtained. However, I have some major concerns:

  1. I am not familiar with urine sample testing, do you need to obtain IRB protocols or other protocols before conducting such experiment? E.g. agreement between university and patients.
  2. The authors mentioned the misuse of antibiotics which lead to antibiotic resistance. Does existence of ampicillin in the urine has direct relationship with the occurrence of antibiotic resistant bacteria? I am assuming urine is sterile... Will correct use of antibiotics not having any drug residues in the urine sample? The antibiotic resistant part seems a little bit not connected with the purpose of the article.
  3. The discussion section is relatively weak. I am specially curious about how currently the antibiotics are detected in the urine sample. I am assuming HPLC-MS is rapid and accurate as well, even with less disturbance from other antibiotics. ELISA might be another strategy. Solid literature review should be conducted.
  4. The novelty of the research. The use of aptamer is interesting but this is published in the processed milk product already. Other than testing in the urine sample, are there any other significant novelties involved?

Reviewer 2 Report

1 “The labelled aptamer strand (10 nM) was annealed at 95°C for 5 min in potassium phosphate buffer containing magnesium chloride (KPi, 100 mM + 1 mM MgCl2, pH 8), before cooling to 4 °C, to ensure the aptamer folds into the correct tertiary structure. The annealed aptamer strands were subsequently incubated with AuNP’s, producing an AuNP-aptamer conjugate with a quenched fluorescence.”

The annealing and cooling step herein might be counter-productive. Without the target, no tertiary structure was needed. The aptamer will fold adaptively with the presence of the target. While with a specific tertiary structure, it is hard to adsorb onto the AuNPs, because the ssDNA in flexible state was more easily to adsorb onto the AuNPs.

And this might be the reason for the high background signal as the authors written that “We note, the high baseline level of fluorescence is believed to be due to incomplete quenching of the FAM-labelled aptamers”.

2 What is the structural difference of the ampicillin and hydrolysed ampicillin? Please add the chemical structures.

3 How is the stability of the developed aptamer-based assay? 

4 How is the repeatability of the developed aptamer-based assay?

5 A comparison of the LOD or LOB with that of other assays for ampicillin should be added.

Reviewer 3 Report

Dear editor,

The submitted work entitled "Aptamer Based Detection of Ampicillin in Urine Samples" uses an already selected aptamer for the detection of Ampicillin via the detachment of Ampicillin fluorescently labeled aptamer from the surface of gold nanoparticles upon target detection. The work can be accepted after major corrections as detailed below:

1) the study does not provide enough control experiments, for example, Figure 2 should be presented along with the response of the sensor to Ampicillin in the presence of a random ssDNA-labelled with the same dye. in addition, the response of the exposing Ampicillin to gold nanoparticles only should be shown.

2) please show what do the error bars represent in all the figures, it could be at the end of the captions

3) please explain what do you mean by "Suitability of FAM Labelled Aptamers" in line 148?

4) the quality of Figure 5 is really bad and should be replaced by a more conventional figure to convey the results of urine fluorescence and its overlap with FAM fluorophore.

5) Again, Figure 6 should show a control sample of the response of random ssDNA to the target, to confirm that the signals arose from specific target-aptamer interactions.

Reviewer 4 Report

The study presented here shown the use of a modify aptamer that was made by another research group (song et al.).

The work is presented well; but a few points are missing.

First of all it does not make any sense the fact that the buffer study was done with an aptamer with FAM attached at one end; while the study with urine sample it has been done with a different fluorescent tag; from FAM you change it to Alexafluor647.

The logic way of doing this will be to use one tag for the buffer and sample study and compare  the two. The authors explain the difference in pH efficiency between the two tag; but this is not enough to justify the change in fluoresencent label between the study.

The authors stated that the calibration curves presented here are linear in the clinical range:

in figure two the curve does not look linear between 1e9 and 1e5; the difference seems to be noticed after a concentration of 1e5. The error bars are extremely high as well; to be able to discern smaller concentrations.

figure 3 show a similar trend; with a flat line between 1e9 and 1e5. Only at higher concentration we seems to see some differences. The figures did not shown a linear calibration curve and no equations or R square value has been presented for any study to justify what the authors stated here.

When looking into the urine samples; it is not clear how many samples have been tested with the methodology described here. three samples for data point seems a small amount to validate your method. A higher number is necessary to support your conclusions.

Round 2

Reviewer 1 Report

I like your feedback regarding the novelty of the research. I would recommend you to put this information somewhere in the discussion to showcase your novelty, in comparison with the previous work.

Author Response

The authors thank the reviewer for their comments and will take their advice on board.

Reviewer 2 Report

The authors said, “While the reviewer may be correct that the aptamer will fold adaptively into the correct 3D conformation in the presence of the target, this has not been confirmed experimentally i.e. it is not known whether the enthalpic gain associated with formation of the aptamer-ampicillin complex willoutcompete the entropic cost. We thus followed protocols used commonly in aptamer-based assays which include an initial annealing phase. We note, the annealing step is common for all experiments, including controls, and therefore the background signal if caused by this would be present in all regardless of the annealing step.”

I still do not agree with the authors. It is well-known that

  • The aptamer will fold adaptively with the presence of the target.
  • ssDNA in flexible state was more easily to adsorb onto the AuNPs.

In addition, the commonly-used annealing step is more suitable for those immobilized aptamers. While in the present one, the annealing and cooling step herein might be counter-productive.

Reviewer 3 Report

The authors addressed the recommendations by reviewers - the manuscript can now be accepted in its present form

Author Response

We thank the reviewer for their time in reviewing the paper.

Reviewer 4 Report

I have reviewed the second  version of this manuscript. The overall descriptions of the techniques used has been improved (number of samples tested has been speficy). I don't agreed with the authors decision of mixing samples from different patient. I understand that the samples were from healthy vounteer, but I would have kept them as separate samples.

No extra work has been done to improve the detection of antibiotics at lower concentration, the authors stated that

"The high limits of detection here are likely due to
inaccuracy at the lower end of the concentration range tested, and the larger variance seen in the sample value, particularly at 10 nM concentrations. "

To improve the methodology that the author are describing in this manuscript I was expecting extra experimental work . The results presented here are not good enough for publications. The method need to work at lower concentration of antibiotics without such large standard deviations.

I would suggest for the authors to spend more time develping the method and resubmit with better results.

Author Response

We strongly disagree with the Reviewers suggestion that the “method need to work at lower concentration of antibiotics”. As discussed in details in the manuscript, the clinically relevant range of ampicillin con centration in urine is between 0.1 and 100 uM and our assay clearly shows Limits of detection well below this, in accordance with the relevant methods set out by IUPAC and commonly used in industrial and academic research for calculations of these.

As with all assays of this type, more results could be collected to reduce errors and we accept the reviewers comments that this would always be more desirable, and would be performed in future work focussed on translation and commercialisation of the assay. Here, we report proof of principle and have employed robust experimental design and statistical assessment that aligns with currently established frameworks used in both industrial and academic research for research and development of molecular assays.

As to the reviewer’s comments on the mixing of urine samples to create our pooled clinical urine samples; pooling of samples preserves donor anonymity and is a stringent requirement for gaining ethical approval for the use and storage of donor samples. If the work is to be taken forward for further trials within a clinical setting this would of course not be the case, and individual’s urine would be kept and tested separately at this point.